# Prostate Cancer-Specific Lysine 53 Acetylation of Cytochrome *c* Drives Metabolic Reprogramming and Protects from Apoptosis in Intact Cells

**DOI:** 10.3390/biom14060695

**Published:** 2024-06-14

**Authors:** Paul T. Morse, Junmei Wan, Tasnim Arroum, Mackenzie K. Herroon, Hasini A. Kalpage, Viktoriia Bazylianska, Icksoo Lee, Elisabeth I. Heath, Izabela Podgorski, Maik Hüttemann

**Affiliations:** 1Center for Molecular Medicine and Genetics, Wayne State University, Detroit, MI 48201, USA; morsepa@wayne.edu (P.T.M.);; 2Department of Pharmacology, Wayne State University, Detroit, MI 48201, USA; 3Department of Biochemistry, Microbiology and Immunology, Wayne State University, Detroit, MI 48201, USA; 4College of Medicine, Dankook University, Cheonan-si 31116, Republic of Korea; icksoolee@dankook.ac.kr; 5Karmanos Cancer Institute, Department of Oncology, Wayne State University, Detroit, MI 48201, USA

**Keywords:** cytochrome *c*, mitochondria, respiration, apoptosis, Warburg effect, prostate cancer, cancer

## Abstract

Cytochrome *c* (Cyt*c*) is important for both mitochondrial respiration and apoptosis, both of which are altered in cancer cells that switch to Warburg metabolism and manage to evade apoptosis. We earlier reported that lysine 53 (K53) of Cyt*c* is acetylated in prostate cancer. K53 is conserved in mammals that is known to be essential for binding to cytochrome *c* oxidase and apoptosis protease activating factor-1 (Apaf-1). Here we report the effects of this acetylation on the main functions of cytochrome *c* by expressing acetylmimetic K53Q in cytochrome *c* double knockout cells. Other cytochrome *c* variants analyzed were wild-type, K53R as a control that maintains the positive charge, and K53I, which is present in some non-mammalian species. Intact cells expressing K53Q cytochrome *c* showed 49% decreased mitochondrial respiration and a concomitant increase in glycolytic activity (Warburg effect). Furthermore, mitochondrial membrane potential was decreased, correlating with notably reduced basal mitochondrial superoxide levels and decreased cell death upon challenge with H_2_O_2_ or staurosporine. To test for markers of cancer aggressiveness and invasiveness, cells were grown in 3D spheroid culture. K53Q cytochrome *c*-expressing cells showed profoundly increased protrusions compared to WT, suggesting increased invasiveness. We propose that K53 acetylation of cytochrome *c* is an adaptive response that mediates prostate cancer metabolic reprogramming and evasion of apoptosis, which are two hallmarks of cancer, to better promote tumor survival and metastasis.

## 1. Introduction

Cytochrome *c* (Cyt*c*) is a globular, 104 amino acid protein that plays a major role in mitochondrial respiration and intrinsic apoptosis, linking these two critical cellular processes. Within the mitochondrial intermembrane space, the primary role of Cyt*c* is to shuttle single electrons from complex III to complex IV (cytochrome *c* oxidase, COX) as part of the electron transport chain (ETC) [1,2]. This enables the process of oxidative phosphorylation (OxPhos), resulting in the production of ATP. Cyt*c* also has other life sustaining functions, including scavenging of reactive oxygen species (ROS) and redox-coupled protein import [3]. During cell death, the release of Cyt*c* from the mitochondria into the cytosol commits the cell to intrinsic apoptosis, highlighting the function of Cyt*c* as either a life-sustaining or pro-apoptotic protein depending on its subcellular localization. In the cytosol, Cyt*c* binds apoptosis protease activating factor-1 (Apaf-1), forming the apoptosome [4]. This activates caspase-9, which activates caspase-3, an effector caspase [5]. However, the sublethal release of Cyt*c* can desensitize the cell to apoptosis [6]. DNA damage has also been shown to induce the translocation of Cyt*c* to the nucleus [7]. Given these crucial processes, Cyt*c* is heavily regulated by ATP, the expression of tissue-specific isoforms, and post-translational modifications [8].

Six tissue-specific phosphorylations, a cancer-specific acetylation, and an acetylation that is gained during ischemia in skeletal muscle have been mapped on Cyt*c*. The phosphorylations are tyrosine 97 and threonine 49 in the normal and aging hearts, respectively [9,10,11], tyrosine 48 in the liver [12,13], threonine 28 and threonine 58 in kidney [14,15], and serine 47 in the brain [16,17]. Generally, these modifications were found to be present under basal conditions in their respective tissues and were lost after ischemia, contributing to ischemia-reperfusion injury [18]. Lysine 39 was identified in porcine tibialis anterior skeletal muscle [19]. It is the first post-translational modification reported on Cyt*c* that is gained during ischemia, increases respiration, and protects the cell from apoptosis.

Initial work pointed to an important role of Cyt*c* as a driver of cancer aggressiveness, but a molecular mechanism at the protein level remained unknown [20]. Lysine 53 (K53) acetylation was identified in prostate cancer (PCa) and was found to promote Warburg metabolism and apoptosis evasion, two hallmarks of cancer [21]. We identified this modification in both prostate cancer xenografts derived from prostate cancer cell lines and in human PCa samples. The modification was not detectable in normal prostate tissue, lung cancer, kidney cancer, or breast cancer. K53 is conserved in mammals and is one of 18 lysines in Cyt*c*, yet it was the only lysine acetylated in prostate cancer, indicating that this modification is PCa-specific. In that initial work, we studied K53 acetylation using acetylmimetic K53Q protein and found that COX activity and caspase-3 activity, representing respiration and intrinsic apoptosis, were 35% and 80% decreased, respectively, in the presence of K53Q Cyt*c*. These findings were expected given that K53 is known to be part of the binding sites on Cyt*c* for both COX and Apaf-1 [4,22]. Furthermore, other functions of Cyt*c* were found to be altered by K53 acetylation. Using acetylmimetic K53Q, the ROS scavenging capability was increased, which would benefit cancer cell survival, while the cardiolipin peroxidase activity was decreased, blocking another pro-apoptotic function of Cyt*c* [23]. The experiments in the initial study were performed using purified protein; however, the effect of K53 acetylation on intact cellular function has not yet been investigated. Given the specificity and regulatory importance of this residue, we now extend our characterization by transfecting Cyt*c* double knockout cells with plasmids coding for recombinant wild-type (WT), acetylmimetic K53Q, K53R as an additional control that maintains the positive charge similar to WT, and nonpolar K53I Cyt*c*, which is present in some non-mammalian species, to investigate the impact of acetylmimetic K53Q replacement on cellular parameters such as metabolism and cell death.

PCa is the most common cancer in men. Additionally, it is the second leading cause of cancer-related deaths among men in the US [24]. In contrast to other cancers, PCa is known to be metabolically heterogeneous [25], yet little is known about specific metabolic vulnerabilities that can be targeted therapeutically. Mitochondrial dysfunction has been shown to correlate with disease aggressiveness [26]. We propose that K53 acetylation of Cyt*c* benefits PCa by facilitating metabolic reprogramming and apoptosis evasion. Our data reported here show that cells expressing acetylmimetic K53Q have decreased mitochondrial respiration with a corresponding increase in glycolytic activity and are protected from cell death. Furthermore, these changes promote more invasive spheroid formation, correlating with a more aggressive phenotype. Altogether, our data indicate that Cyt*c* is a potent driver of PCa.

## 2. Materials and Methods

### 2.1. Site-Directed Mutagenesis, Transfection, and Cell Culture

All chemicals and reagents were purchased from MilliporeSigma (Burlington, MA, USA) unless otherwise specified. A previously generated pBABE-puromycin expression plasmid (Addgene; Cambridge, MA, USA) containing the cDNA sequence for wild-type somatic rodent Cyt*c* (WT) cDNA [14,15,17] was used to generate plasmids for non-acetylated control K53R, acetylmimetic K53Q, and K53I Cyt*c* variants using the QuickChange lightning site-directed mutagenesis kit (Agilent Technologies; Santa Clara, CA, USA) according to the manufacturer’s protocol, as previously described [21]. The following primers were used: K53R forward primer: 5′–CTT ACA CAG ATG CCA ACA GGA ACA AAG GTA TCA CC–3′, K53R reverse primer: 5′–GGT GAT ACC TTT GTT CCT GTT GGC ATC TGT GTA AG–3′, K53Q forward primer: 5′–CTT ACA CAG ATG CCA ACC AGA ACA AAG GTA TCA CC–3′, K53Q reverse primer: 5′–GGT GAT ACC TTT GTT CTG GTT GGC ATC TGT GTA AG–3′, K53I forward primer: 5′–CTT ACA CAG ATG CCA ACA TTA ACA AAG GTATCA CCT G–3′, and K53I reverse primer: 5′–CAG GTG ATA CCT TTG TTA ATG TTG GCA TCT GTG TAA G–3′. As a negative control, an empty vector (EV) plasmid without the sequence for Cyt*c* was also used. After mutagenesis, the final plasmids were confirmed via sequencing. Plasmid constructs for WT, K53R, K53Q, K53I, and EV were stably transfected into Cyt*c* double knockout mouse lung fibroblasts (gifted by Dr. Carlos Moraes, University of Miami, Coral Gables, FL, USA) using Transfast transfection reagent (#E2431, Promega, Madison, WI, USA). The transfected cells were cultured in DMEM growth media (#11965-092, Gibco; Grand Island, NY, USA) supplemented with 10% FBS (#16000-044, Gibco), 100 µg/mL primocin (#ant-pm-1, InvivoGen; San Diego, CA, USA), 1 mM sodium pyruvate (#11360-070, Gibco), and 50 µg/mL uridine (#U3003, MilliporeSigma). Stable cell lines expressing the four recombinant Cyt*c* variants and EV were selected in the presence of 2 µg/mL puromycin and cultured at 37 °C with 5% CO_2_, unless otherwise stated. 

### 2.2. Western Blotting

To determine Cyt*c* expression, cells were lysed using RIPA lysis buffer (150 mM NaCl, 5 mM EDTA, pH 8.0, 50 mM Tris, pH 8.0, 1% NP-40, 0.5% sodium deoxycholate, and 0.1% sodium dodecyl sulfate (SDS)) supplemented with protease inhibitor cocktail (#P8340, MilliporeSigma), sonicated, and centrifuged at 16,900× *g* for 20 min at 4 °C to remove insoluble cell debris. The protein concentration of the lysates was determined using the DC protein assay kit (#5000111, Bio-Rad; Hercules, CA, USA) according to the manufacturer’s protocol. A total of 30 µg of cell lysate from each cell line was run on a 10% tris-tricine SDS-PAGE gel. The gel was transferred onto an immun-blot PVDF membrane (#1620177; Bio-Rad) using a Trans-Blot SD semi-dry apparatus (#1703940; Bio-Rad) at 75 mA for 15 min. The membrane was blocked with 5% milk for 1 h at room temperature. The membrane was incubated in 1:1000 mouse anti-Cyt*c* (#556433; BD Pharmingen; San Jose, CA, USA) in 5% milk overnight at 4 °C. As a loading control, the membrane was incubated in 1:5000 mouse anti-GAPDH (#60004-1-Ig; Proteintech; Rosemont, IL, USA) in 5% milk overnight at 4 °C. For the secondary antibody, the membrane was incubated in 1:10,000 sheep anti-mouse IgG conjugated to horseradish peroxidase secondary antibody (#NA931V; GE Healthcare; Chicago, IL, USA) in 5% milk for 2 h. The blot was visualized using Pierce ECL western blotting substrate (#32106; Thermo Fisher Scientific; Waltham, MA, USA) after a 2-min incubation. 

### 2.3. Oxygen Consumption Rate and Extracellular Acidification Rate 

Cells were seeded at 30,000 cells/well onto a 0.1% gelatin-coated Seahorse XF^e^24 cell culture microplate (#100777-004, Agilent Technologies) and cultured overnight. The next day, growth media was replaced with 675 µL/well Seahorse media (#D5030, MilliporeSigma) supplemented with 10 mM glucose and 10 mM sodium pyruvate without phenol red, FBS, or sodium bicarbonate. Cells were incubated in a CO_2_ free incubator for 1 h to remove dissolved CO_2_ from the media. Oxygen consumption rate (OCR) and extracellular acidification rate (ECAR) were measured in an XF^e^24 Seahorse extracellular flux analyzer. Sequential injections of 1 µM oligomycin, 2.5 µM carbonylcyanide-3-chlorophenylhydrazone (CCCP), and 1 µM rotenone/antimycin A were used for the mitochondrial stress test. The mitochondrial stress test parameters (basal respiration, non-mitochondrial respiration, proton leak, ATP-coupled respiration, maximal respiration, and spare respiratory capacity) were calculated according to the manufacturer’s protocol. The total protein content of each well after the experiment was used to normalize OCR and ECAR values. The OCR is reported as pmol O_2_/min/µg protein, and the ECAR is reported as mpH/min/µg protein.

### 2.4. ATP Production

Cells were seeded at 1 × 10^6^ cells/dish onto 10 cm cell culture dishes (#664160, Greiner Bio-One; Frickenhausen, Germany) and cultured overnight. Within 90 s, cells were collected via scraping and flash frozen. Pellets were stored at −80 °C until analysis. Cell lysis was achieved via boiling for 2 min in 300 µL boiling buffer (100 mM Tris-Cl, 4 mM EDTA, pH 7.75) and sonication on ice. Samples were diluted, and ATP concentration was determined using the ATP bioluminescence assay kit HS II (#11699709001, Roche, Basel, Switzerland), according to the manufacturer’s protocol, using an Optocomp I luminometer (MGM Instruments; Hamden, CT, USA). The total protein content of each sample after the experiment was used to normalize ATP levels. The ATP level is reported as µmol ATP/mg protein.

### 2.5. Growth Rate

Cells were seeded at 40,000 cells/well onto 0.1% gelatin-coated Costar 12-well plates and cultured for up to 4 days. Every 24 h, cells were harvested and counted. 

### 2.6. Mitochondrial Membrane Potential

The relative mitochondrial membrane potentials (ΔΨ_m_) were measured using the ratiometric probe JC-10 (#ENZ-52305, Enzo Life Sciences; Farmingdale, NY, USA). Cells were seeded at 30,000 cells/well onto 0.1% gelatin-coated Costar 96-well plates and cultured overnight. The next day, growth media was replaced with DMEM (#31053-028, Gibco) supplemented with 3 µM JC-10 without phenol red or FBS, and cells were incubated for 30 min at 37 °C. Cells were washed twice with 1× PBS. Green (excitation/emission 485 nm/527 nm) and red (excitation/emission 485 nm/590 nm) fluorescence were measured in 1× PBS using a Synergy H1 microplate reader (BioTek Instruments Inc.; Winooski, VT, USA). ΔΨ_m_ is reported as the ratio of red to green fluorescence.

### 2.7. Mitochondrial Reactive Oxygen Species Production

The mitochondrial ROS production was measured using the mitoSOX probe (#M36008, Invitrogen; Carlsbad, CA, USA). Cells were seeded at 100,000 cells/well onto 0.1% gelatin-coated Costar 24-well plates and cultured overnight. The next day, growth media was replaced with DMEM (#31053-028, Gibco) supplemented with 5 µM mitoSOX without phenol red or FBS, and cells were incubated for 30 min at 37 °C. Cells were washed twice with 1× PBS. Red fluorescence (excitation/emission 510 nm/580 nm) was measured in 1× PBS using a Synergy H1 microplate reader (BioTek Instruments Inc.; Winooski, VT, USA). 

### 2.8. Annexin V/Propidium Iodide Staining and Fluorescence-Activated Cell Sorting

Cell death levels were measured after annexin V and propidium iodide (PI) staining using fluorescence-activated cell sorting analysis. Cells were seeded at 1 × 10^6^ cells/dish onto 10 cm cell culture dishes (#664160, Greiner Bio-One) and cultured overnight. The next day, cells were exposed to either H_2_O_2_ (400 µM for 16 h) or Staurosporine (1 µM for 5 h). Cells were harvested, washed twice with 1× PBS, and resuspended in 1× annexin V binding buffer from the FITC annexin V apoptosis detection kit I (#556547; RRID: AB_2869082; BD Pharmingen). A total of 1 × 10^6^ cells were stained with annexin V and PI. Data were collected on a CyFlow Space flow cytometer (Sysmex America, Inc.; Lincolnshire, IL, USA) and analyzed with FCS Express version 7.0 software (De Novo Software; Glendale, CA, USA). Cell death is reported as a percent of the total cell population.

### 2.9. 3D Culture Assay

Acid-washed coverslips (#CLS-1760-012, Chemglass Life Sciences; Vineland, NJ, USA) were placed in a 35 mm cell culture dish (#430165, Thermo Fisher Scientific; Waltham, MA, USA), and 45 µL of non-diluted (~15 mg/mL) Cultrex (#3445-005-01, R&D Systems; Minneapolis, MN, USA) was gently added on top of each coverslip. The dishes were incubated for 15 min at 37 °C to allow for polymerization. Cells were seeded at 10,000 cells/coverslip and cultured for 60 min at 37 °C to allow for cell adhesion. After the cells were attached, 3 mL of DMEM supplemented with 10% FBS (SH3039603, Cytiva; Marlborough, MA, USA), 2% Cultrex, and 1% penicillin/streptomycin (#15140122, Gibco) was added to each dish. Cells were cultured for 72 h at 37 °C to allow for spheroid growth. Differential interference contrast microscopy images were taken using a Zeiss LSM 780 confocal microscope (Zeiss; Oberkochen, Germany) using both 10× and 40× dipping lenses. ImageJ (National Institutes of Health; Bethesda, MD, USA) was used to quantify the spheroid invasiveness of the 10× images. Using the “freehand selections” tool, the total spheroid area (both the spheroid “body” and invasive spheroid “arms”) was drawn around, and the area was quantified using the “measure” tool. Using the same technique, the spheroid “body” area (excluding invasive spheroid “arms”) was quantified. To determine total invasiveness, the following equation was used: (total area–body area)/body area × 100. Invasiveness is reported as a percentage compared to WT.

### 2.10. Statistical Analyses

The data shown represents the mean. Error bars represent the standard deviation between independent wells or cell culture dishes. Statistical analyses of the data were performed using Graphpad Prism v9.4.1 (Graphpad Software; San Diego, CA, USA). Data were analyzed using a one-way ANOVA, comparing the mean of each column with the mean of every other column with a Tukey post-hoc test. For annexin V/PI experiments, a one-way ANOVA as described above was performed to compare total cell death (PI+, annexin V+, and annexin V+/PI+ combined). *p* values are indicated in the figure legends.

## 3. Results

### 3.1. Generation of Cells Expressing Recombinant WT, K53R, K53Q, and K53I Cytc

To study the effects of K53Q acetylmimetic substitution of Cyt*c* in an intact cellular system, we stably transfected Cyt*c* double knockout cells [27] with WT, non-acetylated control K53R, acetylmimetic K53Q, and nonpolar K53I Cyt*c*. As an additional control, cells were transfected with a plasmid that did not contain the sequence for Cyt*c* to create a Cyt*c*-null empty vector (EV) cell line. While residue 53 is conserved as lysine across mammals and the majority of other species, K53I is the second most common variant found in some non-mammalian species [28]. Most mammals possess a somatic and testes isoform of Cyt*c*, including mice, from which the cell line used is derived. Therefore, the double knockout Cyt*c* system was used to guarantee that knockout of the somatic isoform cannot induce expression of the testes isoform. Clones were selected that equally expressed Cyt*c* to ensure that differences in experimental results were due to Cyt*c* mutation instead of possible differences in Cyt*c* concentration (Figure 1A, Appendix A). 

### 3.2. Expression of K53Q Cytc Results in Decreased Mitochondrial Respiration and Increased Glycolysis

A mitochondrial stress test was performed using a Seahorse bioanalyzer (Figure 1B). The basal oxygen consumption rate (OCR) of K53Q Cyt*c*-expressing cells was 49% decreased compared to the WT expressing cells (Figure 1C). Additionally, the K53Q Cyt*c*-expressing cells showed reduced proton leak, ATP-coupled respiration, maximal respiration, and spare respiratory capacity compared to the WT expressing cells (Appendix A). The rate of glycolysis was indirectly measured using the extracellular acidification rate (ECAR). Showing the opposite behavior compared to OCR, the ECAR of K53Q Cyt*c*-expressing cells was 43% increased compared to the WT-expressing cells, suggesting a metabolic switch from OxPhos to glycolysis (Figure 1D).

### 3.3. Expression of K53Q Cytc Results in Decreased Mitochondrial Membrane Potential and Mitochondrial Reactive Oxygen Species Production

Given the reduced levels of respiration in the K53Q Cyt*c*-expressing cells, we hypothesized that these cells would also show reduced mitochondrial membrane potential (ΔΨ_m_) and ROS production. Respiration, ΔΨ_m_, and mitochondrial ROS production are known to be related [8]. ΔΨ_m_ was measured using JC-10, an improved version of the voltage-dependent JC-1 ratiometric probe. The ΔΨ_m_ of K53Q Cyt*c* expressing cells was reduced compared to the WT-expressing cells, as indicated by a 25% decrease in the ratio of red/green fluorescence (Figure 2A). Similarly, mitochondrial ROS production was measured using MitoSOX. The MitoSOX fluorescence of K53Q Cyt*c*-expressing cells was 71% decreased compared to the WT expressing cells (Figure 2B). Altogether, these data show that the acetylmimetic K53Q substitution leads to an inhibition of respiration, which in turn reduces ΔΨ_m_ mitochondrial ROS production. Additionally, the ATP levels were measured using luminescence produced by an ATP-dependent luciferase catalyzed luciferin oxidation reaction. ATP levels of the K53Q Cyt*c*-expressing cells were 85% decreased compared to the WT-expressing cells (Figure 2C). Additionally, the growth rate of the K53Q Cyt*c*-expressing cells was statistically significantly higher compared to WT-expressing cells (Figure 2D).

### 3.4. Expression of K53Q Cytc Results in Decreased Cell Death after H_2_O_2_ or Staurosporine Treatment

Given the role of Cyt*c* in intrinsic apoptosis, we tested whether K53Q Cyt*c*-expressing cells would show changes in cell death upon challenge, which was measured using annexin V/PI staining followed by fluorescence-activated cell sorting. Cells were treated with either H_2_O_2_ or staurosporine. Annexin V is a marker of early apoptosis, as it binds phosphatidylserine that flips to the outer leaflet of the plasma membrane. PI is a marker of necrosis, as it binds to nuclear DNA. Cell populations that stain for both Annexin V and PI are considered late apoptotic cells [29]. K53Q Cyt*c*-expressing cells showed 15% total cell death after H_2_O_2_ (Figure 3A,C) and 14% total cell death after staurosporine exposure (Figure 3B,D), compared to 27% and 29%, respectively, for WT-expressing cells.

### 3.5. Expression of K53Q Cytc Results in Increased Invasiveness in 3D Cell Culture

Metabolic reprogramming of cancer cells is known to be related to tumor invasiveness and metastases. Given the metabolic changes seen with K53Q-expressing cells, we tested whether there were also changes in invasiveness. Specifically, we used 3D cultures established on a reconstituted basement membrane and monitored the growth and development of invasive outgrowths, consisting of cells migrating out of the spheroid body into the surrounding matrix [30]. The representative 10× and 40× images show clear differences in the size of invasive spheroid arms extruding out of the central spheroid body for the different cell lines (Figure 4A). Strikingly, the K53Q Cyt*c*-expressing cells showed a ten-fold increase in the ratio of invasive spheroid arm area to spheroid body area compared to the WT-expressing cells (Figure 4B), suggesting a significant enhancement in invasive potential.

## 4. Discussion

Cyt*c* has two primary functions: one pro-life as an electron carrier in the ETC, and one pro-death when released into the cytosol to form the apoptosome. Our lab has previously shown that K53 acetylation of Cyt*c* regulates multiple functions of Cyt*c* in vitro, such as COX activity and caspase activity [21]. In that work, we discovered that K53 acetylation of Cyt*c* was present in both castration-sensitive and castration-resistant PCa xenografts, suggesting that the modification is important for PCa regardless of the tumor’s hormone sensitivity. K53 of Cyt*c* is conserved across mammals, highlighting the regulatory importance of this residue [28]. Additionally, this modification was highly detectable in Cyt*c* immunoprecipitated from patient PCa samples and PCa cell lines but not in Cyt*c* from normal prostate, normal kidney, kidney cancer, and lung cancer samples. However, more work is needed to analyze Cyt*c* acetylation in a broader array of cancer specimens to demonstrate its presence in just one or more cancer types.

There are 18 lysine residues in Cyt*c*, making K53 acetylation a PCa-specific modification. Here, we expand that research by characterizing acetylmimetic K53Q Cyt*c* in a cell culture system. Rodents possess two active isoforms of Cyt*c*, referred to as somatic and testes-specific. It is known that knocking out just the somatic isoform induces expression of the testes isoform, necessitating the double knockout system [27]. Humans express only a single, active copy of Cyt*c*, which has features of both the rodent somatic and testes isoforms [31]. After introducing the recombinant Cyt*c* sequences back into the Cyt*c* double knockout fibroblast cells, we examined intact cellular respiration, glycolytic activity, ΔΨ_m_, mitochondrial ROS production, invasiveness, and cell death. We observed that K53 acetylation promotes two cancer hallmarks: Warburg metabolism and apoptosis evasion, as well as increasing the invasive phenotype of the cells in 3D culture. In addition to WT and acetylmimetic K53Q Cyt*c*, we also expressed K53R as an additional control and nonpolar K53I Cyt*c*. As expected, K53R Cyt*c*, which is positively charged similar to non-acetylated WT, functionally behaved like the latter in almost all experiments. We also included K53I Cyt*c*, which is an evolutionarily allowed substitution found in some non-mammalian species, such as the Japanese pufferfish *Takifugu rubripes*, the fruit fly *Drosophila melanogaster*, the sea louse *Caligus clemensi*, some yeasts, and the bacterium *Rhodopila globiformis* [28]. Interestingly, in most experiments, the isoleucine substitution results in functional effects that are in between WT and K53Q Cyt*c*. Because the interaction of Cyt*c* with COX and Apaf-1 is electrostatic in nature, a reduction in binding to its two key partners may explain part of the effect in addition to steric considerations.

K53 lends itself to being an important regulatory residue on Cyt*c* because it directly interacts with both COX [22] and Apaf-1 [4]. The decrease in intact cellular respiration in the acetylmimetic K53Q-Cyt*c* expressing cell line compared to WT was similar to the decrease in COX activity in vitro previously reported using purified K53Q Cyt*c* and COX proteins [21,32]. Furthermore, the glycolytic rate of the acetylmimetic K53Q expressing cell line was increased compared to WT, likely compensating for the reduced mitochondrial respiration. Together, these data support the hypothesis that K53 acetylation helps PCa cells undergo metabolic reprogramming toward Warburg metabolism, a cancer hallmark. For the second cancer hallmark, apoptosis evasion, the decrease in total cell death following H_2_O_2_ or Staurosporine challenge in the acetylmimetic K53Q-Cyt*c* expressing cell line compared to WT was similar to the decrease in caspase-3 activity previously observed using in vitro purified K53Q-Cyt*c* protein in a cell-free system [21]. It is known that the interaction between Cyt*c* and Apaf-1 is mostly electrostatic, with the positively charged Cyt*c* interacting with the negatively charged binding domain on Apaf-1 [33,34]. Given this, it is not surprising that removing a positive charge from Cyt*c* by acetylating K53 or via the acetylmimetic K53Q mutation impacts the ability to bind to Apaf-1 and trigger intrinsic apoptosis. This would facilitate PCa cell survival, even in the event of Cyt*c* release from the mitochondria.

It is well established that mitochondrial respiration, ΔΨ_m_, and mitochondrial ROS production are directly linked [2]. Given the inhibition of mitochondrial respiration seen with acetylmimetic K53Q-expressing cells, we investigated to see if there were changes to ΔΨ_m_, mitochondrial ROS production, and ATP levels. Typically, ΔΨ_m_ is kept between 80 and 120 mV in order to maintain near maximal ATP production while minimizing ROS production [35]. When the ΔΨ_m_ exceeds 140 mV, mitochondrial ROS production increases exponentially [18,36,37]. This relationship was observed here with the acetylmimetic K53Q expressing cells. With the reduction in mitochondrial respiration, we also saw the corresponding decreases in ΔΨ_m_ and mitochondrial ROS production. Interestingly, the ATP levels of the acetylmimetic K53Q expressing cells were strongly reduced. We must note that the Cyt*c* pool in PCa is not fully K53 acetylated in vivo, whereas the acetylmimetic cell culture system used in this study mimics a state of 100% acetylation. Therefore, the difference in ATP levels in PCa in vivo would likely not be as pronounced as observed here.

A weakness of this study is the use of mouse lung fibroblasts. As this is the only Cyt*c* double knockout cell system currently available, future studies should be conducted in a PCa cell system to validate the results seen here.

While treatment for primary PCa has good outcomes, the 5-year survival rate drops precipitously in the event of metastasis [38]. Enhanced glycolysis is a feature of advanced, castration-resistant PCa, where metastases are more common, and is associated with a poor prognosis [39]. We evaluated the metastatic potential of the acetylmimetic K53Q-expressing cells using 3D culture, which revealed dramatic changes in the invasive spheroid extrusion formation. The acetylmimetic K53Q-expressing cells showed about a 10-fold increase in invasive arm formation compared to WT-expressing cells. This suggests that the metabolic reprogramming due to K53 acetylation may contribute to driving prostate tumor metastatic potential.

## 5. Conclusions

Overall, our studies reveal that K53 acetylation is a pro-cancer modification that contributes to metabolic reprogramming and apoptosis evasion (Figure 5). Specifically, this modification inhibits mitochondrial respiration, which points to K53 acetylation playing a progressive role as the PCa phenotype becomes more aggressive. The metabolic activity of PCa is still an active area of research. Some studies suggest that PCa has increased OxPhos activity [40,41,42], while other studies indicate that OxPhos activity eventually decreases at higher stages of the disease [43,44,45]. Given this, a better understanding of K53 Cyt*c* acetylation in PCa is warranted, as the signaling pathway, including acetyltransferases and deacetylases, may provide rational targets to interfere with cancer metabolic reprogramming in addition to re-sensitizing PCa cells to apoptosis.

## Figures and Tables

**Figure 1 biomolecules-14-00695-f001:**
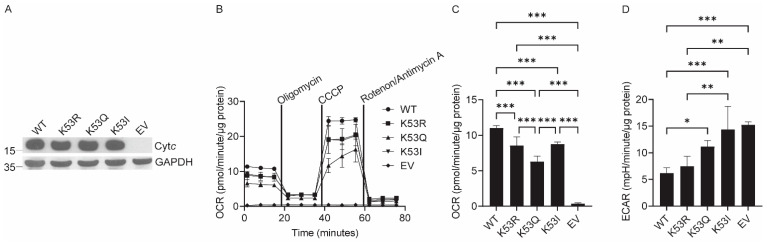
Acetylmimetic K53Q Cyt*c* expressing cells show decreased respiration and increased glycolysis. (**A**) Clones equally expressing Cyt*c* were selected from Cyt*c* double knockout cells stably transfected with WT, K53R, K53Q, and K53I Cyt*c* or EV constructs. (**B**) A mitochondrial stress test was performed in media containing 10 mM glucose and 10 mM sodium pyruvate via sequential injections of 1 µM oligomycin, 2.5 µM carbonylcyanide-3-chlorophenylhydrazone (CCCP), and 1 µM rotenone/antimycin A (*n* = 4–5). (**C**) Basal oxygen consumption rate (OCR) calculated from the mitochondrial stress test (*n* = 4–5). (**D**) Extracellular acidification rate (ECAR) (*n* = 4–5). Data are represented as means ± standard deviation, * *p* < 0.05, ** *p* < 0.01, and *** *p* < 0.001.

**Figure 2 biomolecules-14-00695-f002:**
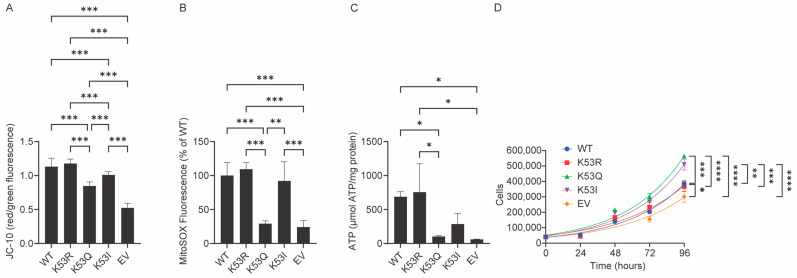
Acetylmimetic K53Q Cyt*c*-expressing cells show reduced mitochondrial membrane potential, mitochondrial ROS production, and ATP. (**A**) The mitochondrial membrane potential of the transfected cells was measured using the ratiometric JC-10 probe and expressed as a ratio of red/green fluorescence (*n* = 14). (**B**) Mitochondrial ROS production of the transfected cells was measured using MitoSOX (*n* = 3–4). (**C**) ATP levels of the transfected cells (*n* = 3). (**D**) Growth rate of the transfected cells (*n* = 3). Data are represented as means ± standard deviation. * *p* < 0.05, ** *p* < 0.01, *** *p* < 0.001, **** *p* < 0.0001.

**Figure 3 biomolecules-14-00695-f003:**
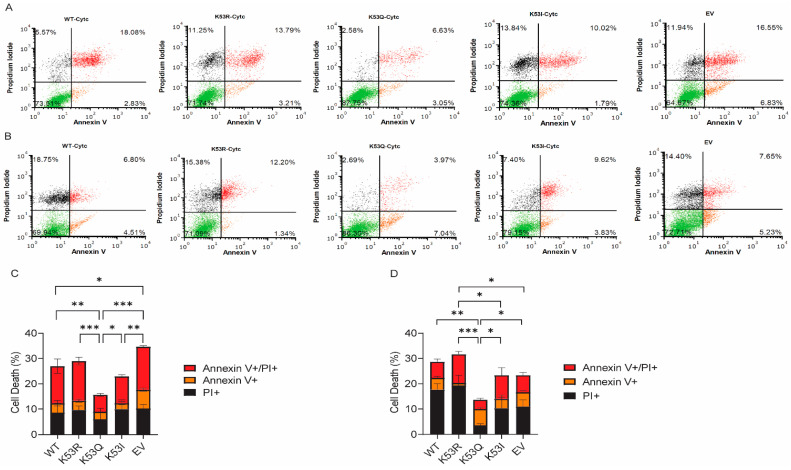
Acetylmimetic K53Q Cyt*c*-expressing cells undergo less cell death after H_2_O_2_ or staurosporine exposure. Representative cell death scatter plots and summary bar charts of transfected cells counting live (green, unstained), necrotic (black, propidium iodide+), early apoptotic (orange, annexin V+), and late apoptotic cells (red, propidium iodide+/annexin V+) after 400 µM H_2_O_2_ for 16 h (**A**,**C**) or 1 µM Staurosporine for 5 h (**B**,**D**) treatment (*n* = 3). Data are represented as means ± standard deviation, * *p* < 0.05, ** *p* < 0.01, and *** *p* < 0.001.

**Figure 4 biomolecules-14-00695-f004:**
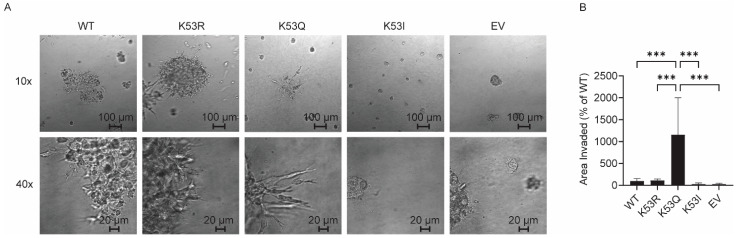
Acetylmimetic K53Q Cyt*c*-expressing cells show increased invasiveness. (**A**) Representative differential interference contrast microscopy images of transfected cells grown in 3D culture at (top) 10× magnification and (bottom) 40× magnification. (**B**) Invasiveness quantitation comparing invasive spheroid arm area over spheroid body area, expressed as a percentage compared to WT (*n* = 6). Data are represented as means ± standard deviation, *** *p* < 0.001.

**Figure 5 biomolecules-14-00695-f005:**
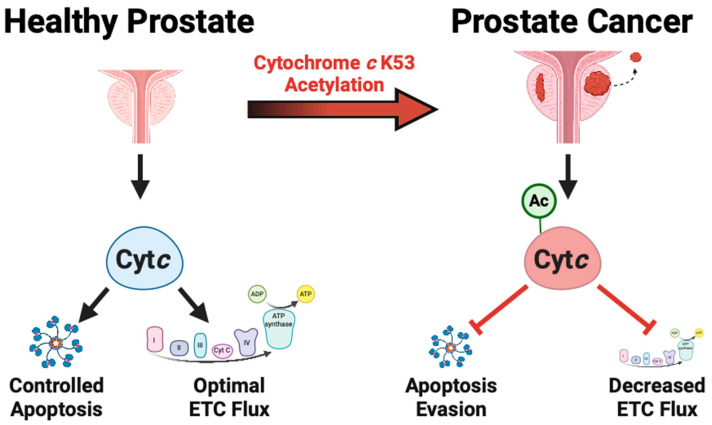
Model of Cyt*c* K53 acetylation as a promoter of metabolic switching (Warburg effect), evasion of apoptosis, and cancer aggressiveness. Created with BioRender.com.

## Data Availability

The original contributions presented in the study are included in the article/Appendix A, further inquiries can be directed to the corresponding author.

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
