# Peer review of "Prostate Cancer-Specific Lysine 53 Acetylation of Cytochrome c Drives Metabolic Reprogramming and Protects from Apoptosis in Intact Cells"

_biomolecules, 2024, doi:10.3390/biom14060695_

Round 1

Reviewer 1 Report

Comments and Suggestions for Authors

In this manuscript Morse and colleagues reported that in comparison to K53R, K53I mutations, cytochrome c acetylmimetic K53Q mutant, which is specifically found in prostate cancer, can promote metabolic reprogramming, evasion of apoptosis and invasion. Using the different stable cells that were generated on basis of CytC double knockout mouse lung fibroblasts, they measured cell glycolysis activity, apoptosis in response to oxidative stress and 3D growth. Although the manuscript overall showed clean data, this is an oversimplified study and there are a number of issues with interpretation of these data.

While it is good strategy to introduce different Cytc mutants in a double knockout system, it undermines the significance and the real biological meaning of these mutants by performing the whole study in a mouse lung fibroblast line. There is no direct evidence to support any information as illustrated in the model. The 3D culture assay does not necessarily measure the cell invasiveness, which should be examined by assays such as invasion chamber or similar.

Reviewer 2 Report

Comments and Suggestions for Authors

Review of Paul T. Morse et al.’s work “Prostate Cancer-Specific Lysine 53 Acetylation of Cytochrome c Drives Metabolic Reprogramming and Protects from Apoptosis

in Intact Cells” 

Introduction 

The article by Paul T. Morse et al. explores the effect of the acetylation of cytochrome c (Cytc) at lysine 53 (K53) on prostate cancer cells. Cytc is a 104-amino acid protein that plays a significant role in cell apoptosis and metabolic pathways. This study aims to investigate the role of the acetylation of Cytc in helping prostate cancer cells survive by avoiding apoptosis and relying on glycolysis for energy. 

The main objective of this article is to explore how the K53 acetylation impacts metabolic pathways such as the ETC and glycolysis along with avoiding apoptosis. The authors hypothesize that K53 acetylation fosters tumor progression. 

The author created cytochrome c variants and introduced them to Cytc double knockout mouse lung fibroblasts. Measured the effect of these changes through techniques such as western blotting, seahorse assay, mitochondrial stress test, ATP assay, JC-10 Probe, MitoSOX Probe, Apoptosis Assay, and Invasiveness Measurement. Overall, the study focuses on investigating the impact of lysine acetylation on cytochrome c and its role in promoting tumor progression in prostate cancer cells.

In short, the acetylation of K53 helps with cancer cell survival and progression through metabolic reprogramming and apoptosis evasion.  

Critical Evaluation

Strengths 

  • The graphs are well-labeled and easy to follow.
  • The authors refer to pre-existing research, demonstrating their comprehensive knowledge of the topic.
  • Used variants of Cytc in the methods making the results more reliable 
  • Impactful study because it considers the impact on two cancer hallmarks.

Weaknesses 

  • Western blotting is a universal technique and could have been explained more concisely with less emphasis. 
  • Did not consider the effects of other possible factors that could promote tumor progression.
  • Could have incorporated research on the role of K53 acetylation in other types of cancer to see if this protein change only impacts prostate cancer cells or all cancer cells in general.
  • Need more research and data on living organisms.

After reviewing the current research on K53 acetylation in prostate cancer cells, it is evident that the role of this protein change extends beyond just prostate cancer. Several studies have indicated the involvement of K53 acetylation in various types of cancer, including breast, lung, and colorectal cancer. Therefore, it would be beneficial to expand the scope of the study to explore the impact of K53 acetylation on other cancer cells as well.

Furthermore, in order to gain a comprehensive understanding of the mechanisms underlying tumor progression, it is essential to consider the potential influence of various factors. It is important to evaluate the interplay between K53 acetylation and other related pathways to determine their collective impact on cancer development and progression.

Moreover, to substantiate the findings and draw more conclusive results, it is imperative to conduct further research and acquire more data on living organisms. Animal models and clinical samples could provide valuable insights into the in vivo relevance of K53 acetylation in cancer progression. This additional data would significantly enhance the relevance and applicability of the research findings.

In a nutshell, this article is impactful and useful since it implies that cytochrome c can be a powerful therapeutic tool to control and treat cancer cells. 

Round 2

Reviewer 1 Report

Comments and Suggestions for Authors

The authors have addressed my comments.